# Pulmonary Ultrasonography in Systemic Sclerosis-Induced Interstitial Lung Disease—A Systematic Review and Meta-Analysis

**DOI:** 10.3390/diagnostics13081429

**Published:** 2023-04-16

**Authors:** Mislav Radić, Hana Đogaš, Andrea Gelemanović, Slavica Jurić Petričević, Ivan Škopljanac, Josipa Radić

**Affiliations:** 1Internal Medicine Department, Rheumatology, Allergology, and Clinical Immunology Division, Center of Excellence for Systemic Sclerosis in Croatia, University Hospital of Split, 21000 Split, Croatia; 2Department of Internal Medicine, School of Medicine, University of Split, 21000 Split, Croatia; josiparadic1973@gmail.com; 3Internal Medicine Department, Nephrology and Haemodialysis Division, University Hospital of Split, 21000 Split, Croatia; hana.dogas@gmail.com; 4Mediterranean Institute for Life Sciences (MedILS), 21000 Split, Croatia; andrea.gelemanovic@gmail.com; 5Pulmonology Department, University Hospital of Split, 21000 Split, Croatia; slavica.juric01@gmail.com (S.J.P.); ivan.skopljanac@gmail.com (I.Š.)

**Keywords:** lung ultrasound, systemic sclerosis, interstitial lung disease, high-resolution computed tomography, systematic review, meta-analysis

## Abstract

Background: The aim of the current systematic review was to summarize and evaluate the overall advantages of lung ultrasonography (LUS) examination using high-resolution computed tomography (HRCT) as a reference standard in assessing the presence of interstitial lung disease (ILD) in systemic sclerosis (SSc) patients. Methods: Databases PubMed, Scopus, and Web of Science were searched for studies evaluating LUSs in ILD assessments including SSc patients on 1 February 2023. In assessing risk of bias and applicability, the Revised Tool for the Quality Assessment of Diagnostic Accuracy Studies (QUADAS-2) was used. A meta-analysis was performed and the mean specificity, sensitivity, and diagnostic odds ratio (DOR) with a 95% confidence interval (CI) were obtained. In addition, in a bivariate meta-analysis, the summary receiver operating characteristic (SROC) curve area was additionally calculated. Results: Nine studies with a total of 888 participants entered the meta-analysis. A meta-analysis was also performed without one study that used pleural irregularity to assess the diagnostic accuracy of LUSs using B-lines (with a total of 868 participants). Overall sensitivity and specificity did not differ significantly, with only the analysis of the B-lines having a specificity of 0.61 (95% CI 0.44–0.85) and a sensitivity of 0.93 (95% CI 0.89–0.98). The diagnostic odds ratio of univariate analysis of the eight studies using the B-lines as a criterion for ILD diagnosis was 45.32 (95% CI 17.88–114.89). The AUC value of the SROC curve was 0.912 (and 0.917 in consideration of all nine studies), which indicates high sensitivity and a low false-positive rate for the majority of the included studies. Conclusions: LUS examination proved to be a valuable tool in discerning which SSc patients should receive additional HRCT scans to detect ILD and therefore reduces the doses of ionizing radiation exposure in SSc patients. However, further studies are needed to achieve consensus in scoring and the evaluation methodology of LUS examination.

## 1. Introduction

Systemic sclerosis (SSc) is an autoimmune disease of the connective tissue that has mainly skin manifestations but sometimes also variable internal-organ involvement and vasculopathy [1]. It is a rare condition, characterized by immune dysregulation and progressive fibrosis, that generally affects mostly young and middle-aged women [2]. Clinical findings that generally present in the early stages of the disease are Raynaud’s phenomenon and gastroesophageal reflux, but as the disease progresses, other, more serious conditions are observed, such as inflammatory skin disease, musculoskeletal inflammation, and fatigue, or in more severe cases, lung fibrosis, pulmonary arterial hypertension, and renal failure. Once SSc is suspected, a definitive diagnosis is made by fulfilling the 2013 European League Against Rheumatism (EULAR) and American College of Rheumatology (ACR) classification criteria [1].

As the disease progresses, the most common cause of death among SSc patients, with a prevalence of 30%, is certainly systemic sclerosis-induced interstitial lung disease (SSc-ILD), with 10-year mortality being as high as 40% [2]. The onset of interstitial lung disease (ILD) is most often within five years of the first non-Raynaud-phenomenon symptom, although when it develops in less than three years, it is considered early-onset and associated with an aggressive clinical course. However, it almost never appears after more than 15 years after a diagnosis of SSc [2].

The golden standard in diagnosis of ILD is the high-resolution computed tomography (HRCT) assessment. The most common radiographic pattern is nonspecific interstitial pneumonia, but other patterns are also seen, such as organizing pneumonia, usual interstitial pneumonia, and pleuroparenchymal fibroelastosis [3]. Various visual methods for quantification of the extent of affected parenchyma have been proposed, but the most frequently used is the Warrick score [4]. Current recommendations state that all patients with SSc should be screened for SSc-ILD using HRCT, it being the primary tool for diagnosing ILD, while pulmonary function tests should be used to support screening and diagnosis [5]. However, HRCT has its disadvantages: higher radiation exposure, especially in frequent scans; cost; and variable availability. Methods of reducing radiation exposure via reducing the number of HRCT slices required to appropriately detect ILD have been evaluated [6].

The need for a quick and easy method of screening has arisen, as there is a high prevalence of lung involvement in early SSc patients, with a strong impact on prognosis—the presence of ILD has a 2.9-fold higher mortality risk [7]. Subsequently, lung ultrasound (LUS) has become a widely explored method given its radiation-free nature and possible bedside application [8]. LUS was proposed for assessment of SSc-ILD, even in its early stages, as it was shown to have a high negative predictive value, according to some studies [7]. The most-analyzed LUS features for ILD assessment are B-lines, although some have suggested B-lines to be too unreliable and subjective. B-lines are defined as vertical and hyperechogenic lines that arise from the pleural line [8].

In the past 5 years, two meta-analyses have been made about LUS evaluation among connective tissue disease (CTD) patients [9,10]. Both studies found that LUSs using B-lines criteria had high diagnostic accuracy and correlated well with HRCT findings, therefore playing an important part in assessing ILD in CTD [9,10]. One analysis also evaluated the optimal number of lung intercostal space (LIS) points to be reviewed in one examination to make a high-accuracy assessment, the number per exam being 14 LIS points [10]. However, no evaluation was made on solely SSc patients, with all previous meta-analyses including all CTD patients. Therefore, the aim of this study is to evaluate the overall advantages of LUS examination using HRCT as a reference standard in assessing the presence and severity of ILD in SSc patients and assess the current criteria used in LUS examination.

## 2. Materials and Methods

### 2.1. Search Strategy

The present study was conducted according to the Preferred Reporting Items for Systematic Reviews and Meta-Analyses for Diagnostic Test Accuracy Studies (PRISMA-DTA) [11]. Using PICO components (population, intervention, comparison, outcome), two researchers (H.Đ. and M.R.) independently performed literature searches. A search of the databases PubMed, Scopus, and Web of Science was performed on 1 February 2023 using the following search phrases: (“lung ultrasound” OR “B lines” OR “pleural irregularity” OR LUS OR “lung ultrasonography” OR “lung sonography”) AND (“pulmonary fibrosis” OR “interstitial lung abnormality” OR “diffuse lung disease” OR “diffuse parenchymal lung disease” OR “interstitial lung disease”) AND (“systemic sclerosis” OR scleroderma). Additional automation filters were applied after the initial search: Review Article, Meeting, Case Report, Editorial Material and Letter filters were used to limit the search, that is, to exclude said categories. Furthermore, a search of reference lists of relevant publications was made to identify additional publications.

### 2.2. Literature Selection

The following inclusion criteria were used:Study design: clinical trials, randomized controlled trials, cohort studies, or observational studies;Studies assessing the presence of interstitial lung disease using lung ultrasound examination compared to HRCT in SSc patients;Studies including human participants over 18 years old;Studies available online and in English;Studies with appropriate data availability.

Therefore, the exclusion criteria were:
Study design: letters, editorials, comments, meeting abstracts, case reports, review, systematic review, or meta-analysis;Studies not assessing the presence of interstitial lung disease using lung ultrasound examination compared to HRCT in SSc patients;Studies including patients with other conditions with impact on pulmonary tissue or connective tissue;Studies with no appropriate data availability or no differentiation of SSc patients among other participants;Studies not including human participants over 18 years old;Studies unavailable online or in English.

### 2.3. Data Extraction

From studies fulfilling the mentioned inclusion criteria, the following data were extracted by two separate researchers (H.Đ and M.R.): author’s surname and year of publication; country of origin; study design; sample size; mean age and disease duration; gender; LUS criteria for evaluation; cut-off points; probe type; numbers of true positives, true negatives, false positives and false negatives; mean total B-lines number; number of lung intercostal spaces examined; and numbers of normal, mild, moderate, and severe cases of ILD according to LUS and HRCT.

### 2.4. Outcome Measures

The primary outcome of this evaluation was to assess the sensitivity and specificity of LUS to detect ILD using HRCT as a reference standard in SSc patients and therefore indicate in what measure and capacity LUS should be used. Secondary outcome measures were to assess the appropriate cut-off values of total B-lines and evaluate the optimal number of LIS points in an examination to achieve a high-quality LUS evaluation.

### 2.5. Quality Assessment

To assess the quality of the included studies, the Revised Tool for the Quality Assessment of Diagnostic Accuracy Studies (QUADAS-2) was used [12]. The QUADAS-2 assesses risk of bias and applicability of studies through four domains of questions: patient selection, index test, reference standard, and flow and timing. Within each domain, risk of bias is regarded as low if all posed questions are answered with a “yes” and as high if one or more posed questions are answered with a “no” or “unclear”. Concern for applicability was regarded as low, high, or unclear in each domain. Regarding overall risk-of-bias evaluation, if in all domains, the risk of bias was low, the overall risk of bias was evaluated as low, while if in one domain, the risk of bias was high or unclear, the overall risk of bias was evaluated as moderate. If in more than one domain, the risk of bias was high or unclear, the overall risk of bias was evaluated as high. Regarding applicability concerns, if one or more domains evaluated indicated a high concern for applicability, the overall concern was regarded as high; otherwise, the concern for applicability was regarded as low.

### 2.6. Statistical Analysis

The meta-analysis was performed in the free software environment for statistical computing, R version 4.0.0 [13], using the meta v6.0-0 [14] and mada v0.5.11 [15] packages. To evaluate the diagnostic accuracy of LUS, two approaches were used. First, a random-effects univariate meta-analysis model was applied, and the mean specificity, sensitivity, and diagnostic odds ratio (DOR) with a 95% confidence interval (CI) were obtained. Heterogeneity between studies was evaluated using the I^2^ statistics, and if the test for heterogeneity was significant (*p*-value < 0.05), it meant that there was significant heterogeneity between the studies. Considering that specificity and sensitivity are mutually connected, in the second phase, a bivariate meta-analysis model was applied, and the summary receiver operating characteristic (SROC) curve area was additionally calculated to examine the overall performance of the diagnostic test (usually, area under the curve (AUC) values higher than 0.8 represent good diagnostic tests). To depict the results of univariate meta-analysis models, forest plots were used, whereas SROC curves were plotted to depict the results of bivariate models.

## 3. Results

### 3.1. Literature Search Results

After an extensive literature search, 161 records were retrieved, among them 37 from PubMed, 80 from Web of Science, and 44 from Scopus. Via applying the Review Article, Meeting, Case Report, Editorial Material and Letter filters, 38 publications were excluded. In addition, 52 duplicates were identified and therefore excluded. The remaining studies’ titles and abstracts were examined, a total of 18 studies were discarded due to study design, and one study was deemed unavailable. One additional study was identified in reference searching. Finally, after a careful, thorough, and independent examination of 53 reports by the two researchers (M.R. and H.Đ.) with all discrepancies resolved by discussion until a consensus was reached, a total of nine studies were included in the present review, as shown in Figure 1.

### 3.2. Characteristics of the Included Studies

In the nine studies included in meta-analysis, all participants were diagnosed with SSc following the American College of Rheumatology/European League Against Rheumatism classification criteria. All participants underwent LUS examination and an HRCT scan simultaneously or within 3 months’ time apart. The summary of basic characteristics is listed in Table 1.

In all studies, LUS examinations were performed by one to a maximum of three experienced ultrasonography practitioners. Eight studies used B-lines as a criterion for diagnosing ILD, while one study used pleural irregularity. Cut-off values for the B-lines varied from more than two in total to more than ten in all areas assessed. The number of LIS points evaluated in an examination varied from a minimum of 10 to all assessed in one exam. All extracted data from the included studies are summarized in Table 2.

### 3.3. Risk-of-Bias and Quality Assessment

Regarding the risk-of-bias assessment, according to the QUADAS-2 evaluation, all studies had low to moderate risk of bias. More specifically, five studies showed low risk of bias, while four studies showed moderate risk of bias, as shown in Table 1. When applicability was assessed, all studies had low concerns. A detailed assessment of each study is shown in Appendix A.

### 3.4. Meta-Analysis of LUS Assessment and B-Line Analysis

Nine studies had available information on TP, TN, FP, and FN and could enter the meta-analysis, with a total of 888 participants. However, since eight studies used B-lines as a criterion for ILD diagnosis [17,18,19,20,21,22,23,24] and one study used pleural irregularity [16], a meta-analysis without the one study using pleural irregularity was performed in order to assess the diagnostic accuracy of LUS (with a total of 868 participants). Due to the significant heterogeneity between the studies (I^2^ = 96% and 63%, respectively), first, a univariate random-effects model was used, and the overall specificity and sensitivity were 0.61 (95% CI 0.44–0.85) and 0.93 (95% CI 0.89–0.98), respectively. Interestingly, the results were not substantially different, even when the one study using pleural irregularity was added into the meta-analysis (specificity = 0.64, sensitivity = 0.94). Forest plots depicting the random-effects meta-analyses of diagnostic accuracy are depicted in Figure 2.

The diagnostic odds ratio (DOR) of the univariate analysis of the eight studies using B-lines as a criterion for ILD diagnosis was 45.32 (95% CI 17.88–114.89), meaning that LUS is 45 times more likely to identify true positives compared to false positives.

Taking into account that the specificity and sensitivity of a diagnostic test are mutually dependent, we additionally performed a bivariate meta-analysis, and the SROC curve is depicted in Figure 3. The AUC value of the SROC curve was 0.912 (and 0.917 in consideration of all nine studies), which indicates high sensitivity and a low false-positive rate for the majority of the included studies.

Five studies [17,18,20,22,24] each used a cut-off of more than five total B-lines in their examinations as an indication of ILD and were therefore used in a subset meta-analysis. The overall specificity and sensitivity from the univariate random-effects model were 0.52 (95% CI 0.33–0.82) and 0.93 (95% CI 0.85–1.02), respectively. The diagnostic odds ratio was 35.05 (95% CI 8.62–142.56) and the AUC value from the bivariate model was 0.876, and even within this subset of five studies, the heterogeneity between them was statistically significant.

### 3.5. Meta-Analysis of the Number of LIS Points Examined

Two studies in this examination used all LIS points, two studies used 72, one study used 58, three studies used 14, and one study used 10 LIS points. To additionally differentiate for this diagnostic criterion, the following subset meta-analyses were performed: a category of four studies evaluating the lower number of examined LIS points (below 15), a category of three studies evaluating the medium number of examined LIS points (between 15 and 72), and finally, the category of the two studies that examined all LIS points. The results of the univariate random-effects and bivariate models are shown in Table 3.

## 4. Discussion

High-resolution computed tomography is the gold-standard method for diagnosing and evaluating the activity of interstitial lung disease [5], but HRCT applies high doses of ionizing radiation, which raises the risk of radiation exposure [6]. Lung ultrasound is a low-cost, noninvasive, and non-ionizing diagnostic technique [25]. Research has demonstrated that LUS using B-line evaluation may be a reliable additional technique for assessing ILD in patients with CTD, including SSc [23,26]. However, its diagnostic value must be confirmed either through large-scale studies or through meta-analysis.

This study represents the first meta-analysis on LUS evaluation in SSc-ILD. In this meta-analysis, we combined evidence on the diagnostic accuracy of lung ultrasound for ILD in patients with SSc. The current meta-analysis of nine studies, with a total of 888 participants, showed that LUS has high diagnostic accuracy. The pooled data showed high overall sensitivity (94%) and somewhat lower specificity (64%), meaning LUS examination very accurately identifies truly negative patients for ILD, while positive findings on LUS should be further evaluated by the golden standard: in this case, HRCT. Even though one study used pleural irregularities as a criterion for diagnosis of ILD while the others used B-lines, in analyzing only the studies using B-lines, the results remained relatively unchanged (sensitivity = 93%, specificity = 61%). These results correlate with the findings of a similar review and meta-analysis conducted on CTD patients [10].

However, even though B-lines are widely used as a LUS criterion in evaluating ILD, it has been suggested by Fairchild et al. that B-lines can be difficult to quantify, are nonspecific, and have a presence that depends on machine settings, frequency, and technique [16]. Pleural irregularities could be easier to visualize and are reproducible while still being strongly associated with the presence of underlying ILD [27]. According to a recent review of LUS in rheumatoid arthritis and autoimmune diseases, both B-lines and pleural-line irregularities have shown significant positive correlations with ILD associated with autoimmune diseases, reaching high sensitivity [28]. Therefore, these concordant results suggest that the choice of which criterion to use in LUS evaluation should be more dependent on the experience and practice of the examining practitioner and the technical specifications and performance of the ultrasound machine used.

There are some crucial points that should be addressed before using LUS as a validated instrument for assessment of ILD-SSc. There is no consensus on how to quantify ILD with LUS—with a dichotomy approach or using quantitative or semiquantitative scoring systems [29]. Lung ultrasonography only evaluates the subpleural regions, whereas HRCT evaluates the complete lung tissue. The vascular bronchial bundle, small nodules dispersed around the bronchial arteries, deep ground glass sign, mediastinal lymph nodes, and thickened and distorted interstitial lesions in deep lung tissue, LUS cannot readily detect [30]. Therefore, as these results have already indicated and because there is no danger of radiation exposure, LUS may be very helpful as an additional method in discerning which patients should proceed to HRCT scanning to confirm ILD diagnosis and in follow-up of patients with ILD-SSc during therapy [30,31].

Furthermore, the cut-off points used for B-lines differed among studies, with the highest number of studies using a cut-off point of more than five total detected B-lines. In the subgroup analysis of these studies, a reduction in specificity (52%) and no significant change in sensitivity (93%) were found compared to the overall result. These results suggest a need to increase the overall cut-off point for a more valid LUS evaluation of ILD. A study by Tardella et al. suggested that the optimal cut-off point would be more than 10 total detected B-lines (with a specificity and a sensitivity each above 90%) [19]. Additionally, a recent study by Gargani et al. also suggested that a screening cut-off value for B-lines should be above 10 total [32]. These findings suggest that most studies apply too-low cut-off values and that there is a need to further explore the benefits of LUS examination in detecting ILD when applying higher cut-off criteria.

Therefore, a growing need of standardizing the scanning procedure for LUS evaluation exists, and it is debated how many intercostal spaces should be examined [10]. A meta-analysis performed on CTD patients suggested that the optimal number of LIS points analyzed (with comparable results of all LIS points analyzed) could be set at 14 and therefore reduce the time needed to perform extensive and thorough evaluation [10]. According to this meta-analysis, analyzing both all LIS points and a lower number of LIS points had high diagnosis values: AUC 0.93 and 0.86, respectively. Therefore, as all-LIS examination could be exceedingly time-consuming and impractical for everyday practice, it has been suggested that a lower number of LIS points examined should be sufficient to maintain the validity of this evaluation. However, as ILD is usually diffuse, a more accurate approach would be a more comprehensive and thorough evaluation, especially for screening SSc patients [33]. In addition, as there was a small number of studies included in this subgroup analysis, it is important to note that these results should be regarded as high-probability rather than high-certainty, considering this as a limitation. Therefore, further studies are needed to assess what would be the optimal number of LIS points evaluated per examination to reduce the impracticality of a time-consuming evaluation.

It is important to note that several studies evaluating the severity of ILD (mild, moderate, or severe category) reported positive correlations between LUS and HRCT categorization [20,22]. Unfortunately, a subgroup meta-analysis was not performed, as there were insufficient data to evaluate the LUS performance on the evaluation of severity of ILD. However, this suggests that LUS examination could have a potential in discerning severity of SSc-ILD and therefore be useful in follow-up examination and therapy.

According to these results, this study is the first to summarize LUS examination quality in detecting ILD in SSc patients. However, there are several limitations in this research that should be addressed. The ultrasound machines and probes used, as well as the scoring method, were significantly different in the included studies. Therefore, this significant interstudy heterogeneity may have had an influence on the acquired results. This indicates that there is a general need to achieve a consensus in LUS methodology, use, scoring, and application in diagnosing SSc-ILD in order to evaluate the benefits of LUS use. Furthermore, the cut-off values and the numbers of LIS points analyzed in LUS examination differed significantly among the studies, which could have also influenced these results. Additionally, this analysis did not include analysis of the value of LUS in discerning severity of ILD due to unavailable data. Severity evaluation could be of significant value in clinical practice. All of these limitations should be considered.

## 5. Conclusions

To summarize, there is a need to standardize LUS protocol in ILD assessment and conduct further large-scale studies to assess the benefits of LUS screening and severity evaluation. LUS examination is a valuable tool in deciding which SSc patients should receive additional HRCT scans to detect ILD and therefore reduces the doses of ionizing radiation exposure in SSc patients. However, the methodologies of LUS examination and scoring differed among the conducted studies. It is imperative that for future clinical applications and better understanding of the value of LUS assessment, a uniform examination and criteria are formed, preferably by professional societies, not only on national levels but on a global scale.

## Figures and Tables

**Figure 1 diagnostics-13-01429-f001:**
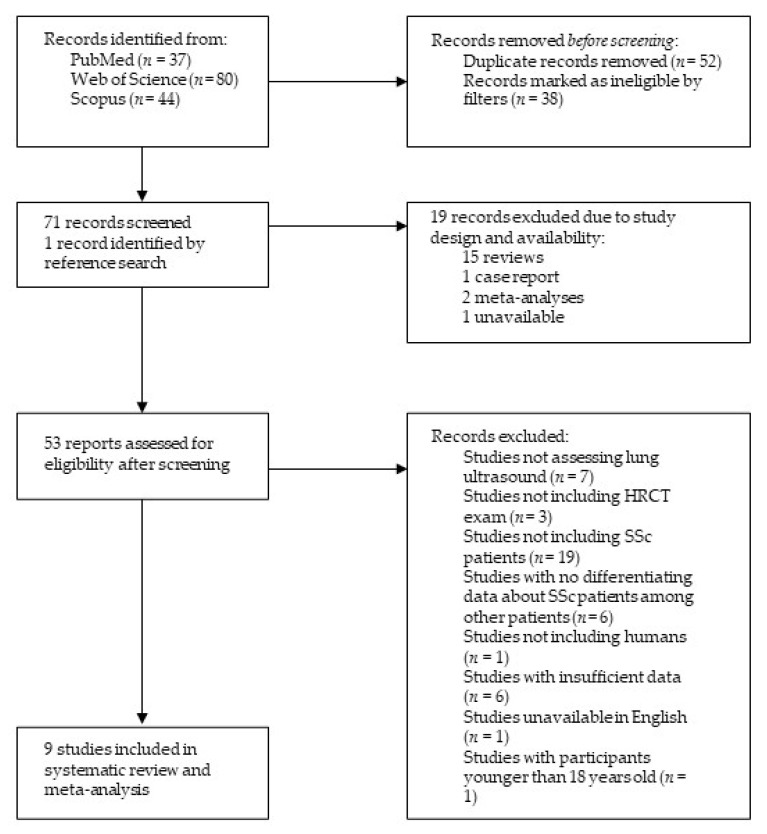
Structural outline of study selection process. Abbreviations: HRCT—high-resolution computed tomography, SSc—systemic sclerosis.

**Figure 2 diagnostics-13-01429-f002:**
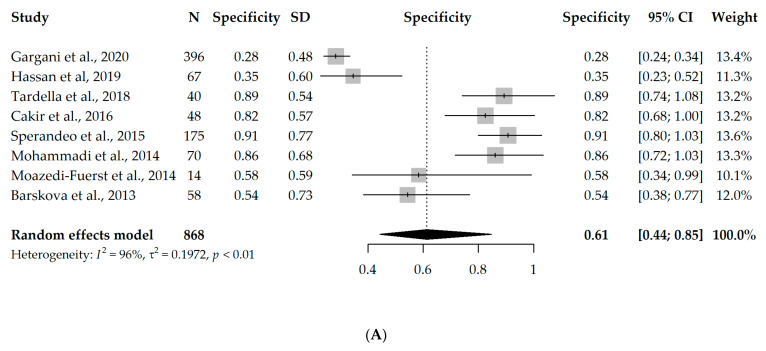
Forest plots for univariate random-effects meta-analyses showing (**A**) specificity and (**B**) sensitivity of LUS, based on eight studies that used B-lines as a criterion for ILD diagnosis [17,18,19,20,21,22,23,24].

**Figure 3 diagnostics-13-01429-f003:**
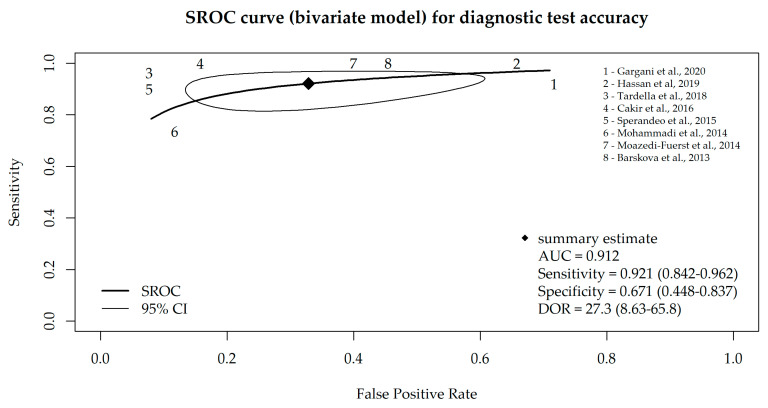
SROC curve for diagnostic test accuracy obtained from the bivariate meta-analysis model of LUS, based on eight studies that used B-lines as a criterion for ILD diagnosis [17,18,19,20,21,22,23,24].

**Table 1 diagnostics-13-01429-t001:** Basic characteristics of included studies.

Author and Year	Country	Sample Size (Total/Male/Female)	Age (Years)	Disease Duration (Years)	QUADAS-2 (Risk of Bias)
Fairchild et al., 2021 [16]	USA	20/NA/NA	NA	NA	Low
Gargani et al., 2020 [17]	Italy	396/31/365	55 (44–66) *	4 (1–9) *	Moderate
Hassan et al., 2019 [18]	Argentina	67/4/63	53 ± 13 **	NA	Low
Tardella et al., 2018 [19]	Italy	40/6/34	56.4 ± 13.42 **	6.5 ± 6.79 **	Low
Çakır et al., 2016 [20]	Turkey	48/2/46	50.8 ± 11.9 **	NA	Moderate
Sperandeo et al., 2015 [21]	Italy	175/9/166	46.46 ± 15.33 **	NA	Low
Mohammadi et al., 2014 [22]	USA	70/8/62	50.29 ± 9.7 **	7.33 ± 6.93 **	Moderate
Moazedi-Fuerst et al., 2014 [23]	Austria	14/NA/NA	NA	NA	Moderate
Barskova et al., 2013 [24]	Italy	58/4/54	51 ± 14 **	NA	Low

* Median (IQR), ** Mean ± SD, Abbreviations: QUADAS-2—Revised Tool for the Quality Assessment of Diagnostic Accuracy Studies, USA—United States of America, NA—not applicable.

**Table 2 diagnostics-13-01429-t002:** Data for performing LUS examinations.

Author and Year	Number of LIS Points Assessed	LUS Criterion Used	Cut-Off Values (Total B-Lines)	Probe	TP	FP	TN	FN
Fairchild et al., 2021 [16]	14	Pleural thickening and granularity, pleural irregularity	NA	Medium-frequency linear probe	11	1	8	0
Gargani et al., 2020 [17]	58	B-lines	≥5	2.5–3.5 MHz cardiac-sector transducers	46	248	98	4
Hassan et al., 2019 [18]	72	B-lines	≥5	Convex transducer of 3.5 MHz	29	25	13	0
Tardella et al., 2018 [19]	14	B-lines	>10	4–13 MHz broadband linear transducer	26	1	12	1
Çakır et al., 2016 [20]	14	B-lines	>5	5 to 10 MHz linear probe	29	3	16	0
Sperandeo et al., 2015 [21]	All	B-lines	>3	3.5–5-MHz convex probe	134	2	24	15
Mohammadi et al., 2014 [22]	10	B-lines	>5	Broadband linear multifrequency transducer of 7–10 MHz	39	2	15	14
Moazedi-Fuerst et al., 2014 [23]	All	B-lines	>2	3.5 MHz convex transducer	9	2	3	0
Barskova et al., 2013 [24]	72	B-lines	>5	2.5–3.5 MHz cardiac sector transducer	36	10	12	0

Abbreviations: LIS—lung intercostal space, LUS—lung ultrasound, TP—true positive, FP—false positive, TN—true negative, FN—false negative, NA—not applicable.

**Table 3 diagnostics-13-01429-t003:** Meta-analysis of three LIS subgroups analyzed.

	Univariate Model		Bivariate Model
	Diagnostic Accuracy (95% CI)	I^2^ (*p*-Value)	Diagnostic Accuracy (95% CI)
**Category of Four Studies Evaluating the Lower Number of Examined LIS Points (Below 15)**			
Specificity	0.86 (0.78–0.95)	0% (0.95)	0.853 (0.74–0.922)
Sensitivity	0.91 (0.80–1.03)	74% (<0.01)	0.916 (0.713–0.98)
DOR	97.49 (29.67–320.38)	100% (<0.001)	88.7 (12.7–315)
AUC	/		0.867
**Category of Three Studies Evaluating the Medium Number of Examined LIS Points (Between 15 and 72)**			
Specificity	0.37 (0.25–0.55)	82% (<0.01)	0.37 (0.246–0.513)
Sensitivity	0.97 (0.93–1.01)	15% (0.31)	0.965 (0.856–0.992)
DOR	22.40 (3.86–129.79)	100% (<0.01)	27.5 (2.15–125)
AUC	/		0.686
**Category of Two Studies That Examined All LIS Points**			
Specificity	0.79 (0.52–1.18)	60% (0.11)	0.797 (0.364–0.964)
Sensitivity	0.90 (0.86–0.95)	0% (0.54)	0.921 (0.826–0.966)
DOR	47.64 (15.25–148.80)	100% (<0.01)	61.1 (10.2–204)
AUC	/		0.934

## Data Availability

The data presented in this study are available within this article and in its Appendix A.

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
