# Peer review of "Pulmonary Ultrasonography in Systemic Sclerosis-Induced Interstitial Lung Disease—A Systematic Review and Meta-Analysis"

_diagnostics, 2023, doi:10.3390/diagnostics13081429_

Round 1
Reviewer 1 Report
The following must be considered in the revision of MS:
The MS under review is Pulmonary ultrasonography in systemic sclerosis-induced interstitial lung disease – a systematic review and meta-analysis
The author must highlight key finding at least three to five in a separate heading.
What are the limitations of the study designed must be presented in a short para.
The structural outline, which is mentioned in figure 1. A similar box must be included as done in the case of record excluded. At the start of the figure 1, include a box ( with detial such as study design: clinical trials, randomized controlled trials, cohort studies or observational studies, studies assessing the presence of interstitial lung disease using lung ultrasound examination compared to HRCT in SSc patients, studies including human participants over 18 years old, studies available online and in English, studies with appropriate data availability).
Minor language correction is recommended.
A more comprehensive flow is highly recommended, as some of the information are missing, in the figure 1.
Author Response
Response to Reviewer 1
We express our sincere gratitude in dedicating your time to review our submitted manuscript. We hope that in our correction we succeeded in fulfilling the requests to your satisfaction.
1) The author must highlight key finding at least three to five in a separate heading.
Following the journal instructions for authors we added the conclusion section in which we highlighted key findings and it now states:
“5. Conclusion
To summarize everything stated, there is a need to standardize the LUS protocol in ILD assessment and conduct further large-scale studies to assess the benefits of LUS screening and severity evaluation. LUS examination is a valuable tool in deciding which SSc patients should receive additional HRCT scan to detect ILD and therefore reduces the dose of ionizing radiation exposure in SSc patients. However, the methodology of LUS examination and scoring differs among the conducted studies. It is imperative that for future clinical application and better understanding of the value of LUS assessment a uniform examination and criteria are formed, preferably by professional societies not only on national levels but on a global scale.”
2) What are the limitations of the study designed must be presented in a short para.
We isolated and fulfilled the short para with a few more limitations and it now states:
“According to these results, this study is the first to summarize LUS examination quality in detecting ILD in SSc patients. However, there are several limitations in this re-search that should be addressed. The ultrasound machines and probes used, as well as the scoring method, are significantly different in included studies. Therefore, this signifi-cant inter-study heterogeneity may have had influence on acquired results. This indicates that there is a general need to achieve a consensus in LUS methodology, use, scoring and application in diagnosing SSc-ILD in order to evaluate the benefits of LUS use. Further-more, cut-off values and number of LIS points analyzed in LUS examination differ signif-icantly among studies which could have also influenced these results. Additionally, this analysis did not include the analysis of the value of LUS in discerning the severity of ILD due to unavailable data. The severity evaluation could be of significant value in clinical practice. All of these limitations should be considered.”
3) The structural outline, which is mentioned in figure 1. A similar box must be included as done in the case of record excluded. At the start of the figure 1, include a box (with detial such as study design: clinical trials, randomized controlled trials, cohort studies or observational studies, studies assessing the presence of interstitial lung disease using lung ultrasound examination compared to HRCT in SSc patients, studies including human participants over 18 years old, studies available online and in English, studies with appropriate data availability).
A more comprehensive flow is highly recommended, as some of the information are missing, in the figure 1.
We reviewed the studies and numbers where we caught few miscalculations and typos which we corrected. We hope we added the necessary details and made the flow more comprehensible. The Figure 1. and text now state:
“3.1. Literature search results
After an extensive literature search 161 records were retrieved, among them 37 from PubMed, 80 from Web of Science and 44 from Scopus. By applying the Review Article, Meeting, Case report, Editorial Material and Letter filters, 38 publications were excluded. Also, 52 duplicates were identified and therefore excluded. The remaining studies titles and abstracts were examined and a total of 18 studies were discarded due to study design and one study was deemed unavailable. One additional study was identified in reference searching. Finally, after a careful, thorough and independent examination of 53 reports by the two authors (M.R. and H.Đ.) with all discrepancies resolved by discussion until a consensus was reached, a total of nine studies were included in present review as shown in Figure 1.”

Reviewer 2 Report
Thank you for inviting me to review this interesting article. In the past decade, the role of lung ultrasound in the management of CTD-ILD is highlighted gradually, especially in scleroderma and rheumatoid arthritis. High sensitivity, feasibility, as well as radiation-free make lung ultrasound more suitable for screening and follow-up of ILD patients, even at the subclinical stage. However, the standardization and validation of LUS examination in CTD-ILD have not yet been established. Different scoring methods and probe frequency are used in clinical operation. In addition, the calculation of the number of B-lines depends on the subjective judgement of the operator. These aforementioned shortcomings lead to high heterogeneity of involved studies, attenuate the statistical power. The authors must especially emphasize in the context, and put forward a solution.
In addition, “Heterogeneity between studies was evaluated using the I2 statistics, and if the test for heterogeneity was significant (p-value<0.05) it meant that there was significant heterogeneity between the studies.”My opinion is that used “p-value<0.05” to test heterogeneity is a little rough. I2 value can be divided into ranges of 25%, 50% and 75% to judge the degree of the heterogeneity and determine the source of the heterogeneity. If the heterogeneity is too large (I2>75% or more), not necessarily suitable for quantitative meta-analysis.
The conversion of data obtained from the original study is not clearly explained. For example, specificity and sensitivity in the original study should be attached to their accuracy by means and confidence interval, but how to convert them into mean and standard deviation in FIG. 2 should be explained.
Author Response
Response to Reviewer 2
We express our sincere gratitude in dedicating your time to review our submitted manuscript. We hope that in our correction we succeeded in fulfilling the requests to your satisfaction.
1) High sensitivity, feasibility, as well as radiation-free make lung ultrasound more suitable for screening and follow-up of ILD patients, even at the subclinical stage. However, the standardization and validation of LUS examination in CTD-ILD have not yet been established. Different scoring methods and probe frequency are used in clinical operation. In addition, the calculation of the number of B-lines depends on the subjective judgement of the operator. These aforementioned shortcomings lead to high heterogeneity of involved studies, attenuate the statistical power. The authors must especially emphasize in the context, and put forward a solution.
To summarize and highlight the need of LUS standardization we added a conclusion section which now states:
“5. Conclusion
To summarize everything stated, there is a need to standardize the LUS protocol in ILD assessment and conduct further large-scale studies to assess the benefits of LUS screening and severity evaluation. LUS examination is a valuable tool in deciding which SSc patients should receive additional HRCT scan to detect ILD and therefore reduces the dose of ionizing radiation exposure in SSc patients. However, the methodology of LUS examination and scoring differs among the conducted studies. It is imperative that for future clinical application and better understanding of the value of LUS assessment a uniform examination and criteria are formed, preferably by professional societies not only on national levels but on a global scale.”
2) In addition, “Heterogeneity between studies was evaluated using the I2 statistics, and if the test for heterogeneity was significant (p-value<0.05) it meant that there was significant heterogeneity between the studies.”My opinion is that used “p-value<0.05” to test heterogeneity is a little rough. I2 value can be divided into ranges of 25%, 50% and 75% to judge the degree of the heterogeneity and determine the source of the heterogeneity. If the heterogeneity is too large (I2>75% or more), not necessarily suitable for quantitative meta-analysis.
Thank you, we fully agree with your comment and that is why we provided both I2 statistics and its accompanying p-value for heterogeneity. We merely use it to check how heterogeneous the studies are, not to define based on this heterogeneity measure should the meta-analysis be performed or not. For situations where p-value is less than 0.05 and we have I2 statistics of 63% or 74% (other situations are clear) these can be debatable if they suffer from significant heterogeneity or some moderate, but exactly because of this kind of interpretation we leave both I2 and p-value.
3) The conversion of data obtained from the original study is not clearly explained. For example, specificity and sensitivity in the original study should be attached to their accuracy by means and confidence interval, but how to convert them into mean and standard deviation in FIG. 2 should be explained.
Thank you for your comment, we noticed that our figures had misleading labels ("mean); we changed the Figure 2 and column names in the Table 1 to match what is actually depicted on them. Specificity and sensitivity are calculated from TP, FP, TN and FN values using the mada R package as described in methodology. Values in the figures and table actually represent the exact sensitivity/specificity calculated for each study. This package provides 95% CI with diagnostic accuracies, and from these values we calculated standard error and standard deviation to be able to make the figure according to the pipeline from meta package to make forest plots.
Conversions from 95% CI to SE and SD are followed using the traditional formulas; SE = (upper limit - lower limit) / 3.92 ; SD = SE x sqrt(N)
We apologise for misleading labels, everything is corrected.

Reviewer 3 Report
In this systematic review and meta-analysis, Radic et al aimed to estimate the value of lung ultrasound in patients with SSc ILD.
The manuscript follows the PRISMA guidelines and is undoubtedly important.
However, there are some issues that need to be addressed:
1. There is inconsistency in the PRISMA flow-chart numbers and between the chart itself and the main text. Thus, the study selection may be flawed.
2. Has the full search strategy been disclosed? I would also use other keywords such as "thoracic ultrasound", "lung involvement" and severl other variants that could be missing.
This could have led to the exclusion of important works such as:
Buda N, Piskunowicz M, Porzezińska M, Kosiak W, Zdrojewski Z. Lung ultrasonography in the evaluation of interstitial lung disease in systemic connective tissue diseases: criteria and severity of pulmonary fibrosis – analysis of 52 patients. Ultraschall Med 2016;37:379–85
Aghdashi M, Broofeh B, Mohammadi A. Diagnostic performances of high resolution trans-thoracic lung ultrasonography in pulmonary alveoli-interstitial involvement of rheumatoid lung disease. Int J Clin Exp Med 2013;6:562-6
Gutierrez M, Salaffi F, Carotti M, Tardella M, Pineda C, Bertolazzi C, et al. Utility of a simplified ultrasound assessment to assess interstitial pulmonary fibrosis in connective tissue disorders— preliminary results. Arthritis Res Ther 2011;13:R134.
Author Response
Response to Reviewer 3
We express our sincere gratitude in dedicating your time to review our submitted manuscript. We hope that in our correction we succeeded in fulfilling all requests to your satisfaction.
- There is inconsistency in the PRISMA flow-chart numbers and between the chart itself and the main text. Thus, the study selection may be flawed.
We sincerely apologise, in our review we caught a few miscalculations and typos which we corrected. The numbers were reviewed all as were the studies appearing in our search strategy and now state:
“3.1. Literature search results
After an extensive literature search 161 records were retrieved, among them 37 from PubMed, 80 from Web of Science and 44 from Scopus. By applying the Review Article, Meeting, Case report, Editorial Material and Letter filters, 38 publications were excluded. Also, 52 duplicates were identified and therefore excluded. The remaining studies titles and abstracts were examined and a total of 18 studies were discarded due to study design and one study was deemed unavailable. One additional study was identified in reference searching. Finally, after a careful, thorough and independent examination of 53 reports by the two authors (M.R. and H.Đ.) with all discrepancies resolved by discussion until a consensus was reached, a total of nine studies were included in present review as shown in Figure 1.”
- Has the full search strategy been disclosed? I would also use other keywords such as "thoracic ultrasound", "lung involvement" and severl other variants that could be missing.
We reevaluated our strategy and no additional research was found that fit our inclusion and exclusion criteria. In fact, we did identify these important works that were excluded as follows:
- Buda N, Piskunowicz M, Porzezińska M, Kosiak W, Zdrojewski Z. Lung ultrasonography in the evaluation of interstitial lung disease in systemic connective tissue diseases: criteria and severity of pulmonary fibrosis – analysis of 52 patients. Ultraschall Med 2016; 37:379–85
Although undoubtedly important, this article does not meet the inclusion criteria, that is, it does not specify how many patients had systemic sclerosis, and within that, it does not provide sufficient data about true positives, false positives, true negatives, and false negatives regarding lung ultrasound examination for detecting interstitial lung disease in specifically only systemic sclerosis patients and therefore cannot be included in our research. In our search, we made an effort to include as many studies as we could by a detailed examination of articles similar to this one that we found with our search strategy to try and extract data for only systemic sclerosis patients.
- Aghdashi M, Broofeh B, Mohammadi A. Diagnostic performances of high resolution transthoracic lung ultrasonography in pulmonary alveoli-interstitial involvement of rheumatoid lung disease. Int J Clin Exp Med 2013; 6:562-6
This study, although important, does not specify the inclusion of only systemic sclerosis patients nor it differentiates the data for only systemic sclerosis patients, it included also patients with other connective tissue disease. Furthermore, it cannot be included due to not reporting full data needed for analysis – true positives, true negatives, false positives and false negatives.
- Gutierrez M, Salaffi F, Carotti M, Tardella M, Pineda C, Bertolazzi C, et al. Utility of a simplified ultrasound assessment to assess interstitial pulmonary fibrosis in connective tissue disorders— preliminary results. Arthritis Res Ther 2011;13: R134.
Again, due to the inclusion of only systemic sclerosis patients this study does not meet inclusion criteria, that is, it does not differentiate the results for only systemic sclerosis patients. All studies with a similar topic were included if they differentiated the data provided for only systemic sclerosis patients.
The reason why we included only systemic sclerosis patients is because our institution, University Hospital of Split, is also a Referral Centre for Systemic Sclerosis and this work presents an important part of our clinical practice. We wanted to explore how can we improve clinical care and diagnostics for this specific population. We hope we resolved all discrepancies to your satisfaction.

Round 2
Reviewer 1 Report
Recommended, English language and style are fine/minor spell check required
Reviewer 3 Report
The authors addressed the suggestions. I have no further comments.